# EML4-ALK Gene Mutation Detected with New NGS Lung Cancer Panel CDx Using Sputum Cytology in a Case of Advanced NSCLC

**DOI:** 10.3390/diagnostics13142327

**Published:** 2023-07-10

**Authors:** Kei Morikawa, Kohei Kinoshita, Shin Matsuzawa, Hirotaka Kida, Hiroshi Handa, Takeo Inoue, Seiji Nakamura, Yoshiharu Sato, Masamichi Mineshita

**Affiliations:** 1Division of Respiratory Diseases, Department of Internal Medicine, St. Marianna University School of Medicine, Kawasaki 216-8511, Japan; kino1989@marianna-u.ac.jp (K.K.); s2matsuzawa@marianna-u.ac (S.M.); h2kida@marianna-u.ac.jp (H.K.); hiroshihstv@marianna-u.ac.jp (H.H.); t2inoue@marianna-u.ac.jp (T.I.); m-mine@marianna-u.ac.jp (M.M.); 2DNA Chip Research Inc., Tokyo 105-0022, Japan; nakamura@dna-chip.co.jp (S.N.); yo-sato@dna-chip.co.jp (Y.S.)

**Keywords:** EML4-ALK lung adenocarcinoma, lung cancer compact panel TM, next-generation sequencing, non-small cell lung cancer, sputum cytology

## Abstract

The detection of driver gene mutations has become essential for lung cancer; however, insufficient sample sizes make gene panel tests difficult to use. We previously reported that the lung cancer compact panel TM (LCCP) could detect EGFR and MET gene mutations with sputum cytology. To date, the detection of gene mutation using RNA from sputum samples is considered practically difficult. We report a case in which the EML4-ALK fusion gene was successfully detected from a sputum sample using the LCCP that was just released in Japan as a new next-generation sequencing lung cancer panel, CDx.

**Figure 1 diagnostics-13-02327-f001:**
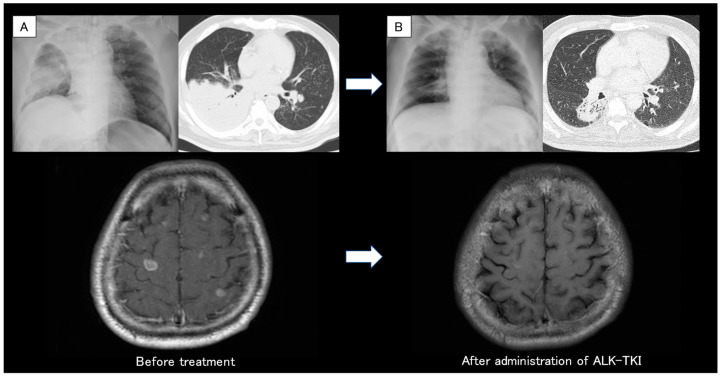
(**A**) Chest X-ray and head CT revealed the primary lung cancer at the right lower lobe with upper lobe atelectasis and multiple brain metastases. (**B**) Chest and head CT showed a favorable response after administration of molecular-targeted drugs for one month. CT: computed tomography. In the diagnosis of non-small cell lung cancer (NSCLC), the detection of driver gene mutations has become an essential test for treatment decisions [1]. On the other hand, as gene mutation panel tests need high-quality and adequate tissue samples, it can often be difficult to perform a gene batch analysis [2]. We reported that the lung cancer compact panel TM (LCCP), which was recently approved for use in Japan, was successful in detecting cases with epidermal growth factor receptor (EGFR) and MET gene mutations using sputum cytology [3]. However, as the quality of RNA tends to degrade quickly, it is generally recommended to use frozen or formalin-fixed, paraffin-embedded (FFPE) tissue specimens to secure a sufficient amount of nucleic acids. Conversely, if high-quality nucleic acid storage is available and high-sensitivity next-generation sequencing (NGS) can be implemented, it is possible to analyze RNA from cytological specimens. We report a case in which the echinoderm microtubule-associated, protein-like 4 anaplastic lymphoma kinase (EML4-ALK) fusion gene was detected in a sputum cytological specimen using LCCP (UMIN000047215/HREC ID 4814). This article is part of a prospective observational study of gene panel analysis using cytological specimens that is currently underway for patients with suspected lung malignancies. Cytological specimens such as transbronchial biopsy (TBB) brushing, transbronchial needle aspiration (TBNA), and pleural effusion were used for genetic panel testing. This study is ongoing since May 2020 (HREC ID 4814). Sputum was also an accepted sample in this study. Sputum and cytological specimens were collected in a sample container (GM tube, Genemetrics, Osaka, Japan) which contained a nucleic acid stabilizer, and they were processed with the Maxwell^®^ RSC Blood DNA Kit and Maxwell^®^ RSC Simply RNA Cells Kit (Promega, WI, USA). Using the purified nucleic acid, the LCCP TM (DNA Chip Research Inc., Tokyo, Japan) NGS assay was performed. The LCCP TM is an amplicon-based high-sensitivity NGS panel capable of measuring eight druggable genes (EGFR, BRAF, KRAS, ERBB2, ALK, ROS1, MET, and RET) for lung cancer. The LCCP was recently approved for clinical practice in Japan as a companion diagnostic kit. Analytical performance, clinical validity, and details of the analysis of the LCCP were previously reported [4]. The patient was a 67-year-old man who had never smoked but had a history of hypertension and type 2 diabetes. A family member found him collapsed in his bathroom, and an ambulance was called. Generalized tonic convulsion was also present during emergency transport. After arriving at the hospital, endotracheal intubation and mechanical ventilation were performed. The CT revealed multiple brain metastases (12 mm or less, 20 or more sites), a mass of more than 10 cm in the lower right lobe, and atelectasis in the upper right lobe (**A**). It was presumed that the seizure was associated with the brain metastases from the primary lung cancer. Tumor markers were CEA 18.8 ng/mL, SLX 1160 U/mL, CYFRA 11.2 ng/mL, SCC 4.2 ng/mL, ProGRP 74.5 pg/mL, and NSE 39.5 ng/mL. Since his condition stabilized with anticonvulsants and sedatives, he was extubated on hospital day 4 and was transferred to our respiratory medicine department. The next day, the sputum that he coughed up was cytologically positive for class Ⅴ adenocarcinoma (Figure 2A), and EGFR cobas^®^ (liquid) was negative. The primary right lower lobe lung adenocarcinoma was considered to be T4N3M1c and c-stage IVB. A CT-guided biopsy was performed on day 7 for tissue sampling (Figure 2B). Whole-brain irradiation was initiated on the same day for multiple symptomatic brain metastases (30 Gy/10 Fr). NGS analysis (LCCP) of the CT-guided biopsy tissue revealed EML4-ALK positive on day 15, and immunohistochemical (IHC) staining/fluorescence in situ hybridization (FISH) confirmed each positive finding (Figure 2C,D). First-line brigatinib of 90 mg/day was administered on day 18, and the dose was increased to 180 mg/day after one week. Both the primary tumor and brain metastases showed remarkable improvements after one month (**B**) with CT assessment. Sputum collected on day 4 for LCCP analysis was successful in the detection of the EML4-ALK fusion gene (Figure 3). The detected RNA mutational variants extracted from tissues and sputum matched EML4exon6ins33_ALKexon20.

**Figure 2 diagnostics-13-02327-f002:**
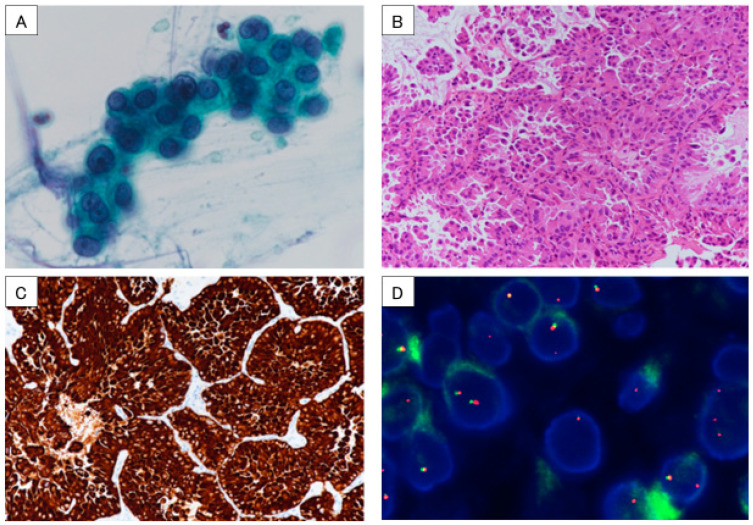
Sputum cytology collected on day 4 of hospitalization (**A**). Pathological image of CT-guided biopsy tissue revealed adenocarcinoma (**B**) and IHC (**C**); FISH (**D**) confirmed both ALK positives. CT: computed tomography, IHC: immunohistochemical, FISH: fluorescence in situ hybridization, ALK: anaplastic lymphoma kinase.

**Figure 3 diagnostics-13-02327-f003:**
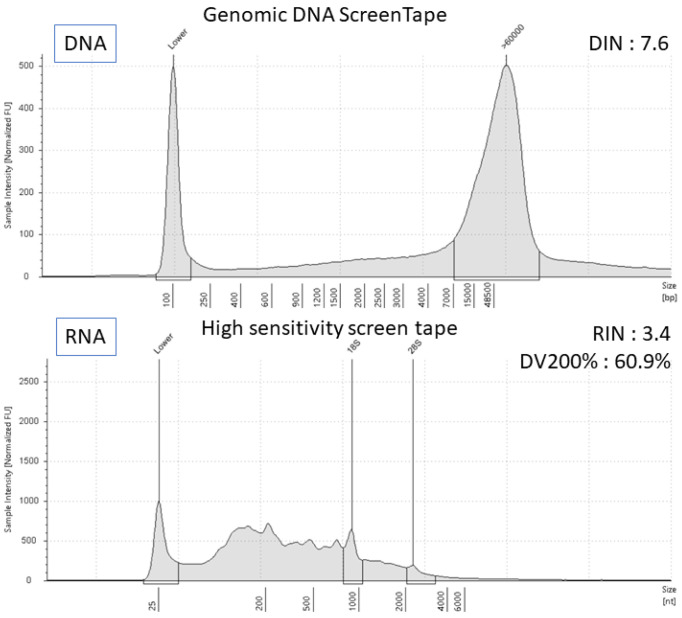
The quality of DNA and RNA purified from sputum samples was analyzed using the Agilent Genomic DNA ScreenTape assay and the High Sensitivity RNA ScreenTape assay of Agilent TapeStation systems. The integrities of DNA and RNA were assessed with the DIN score and RINe score, respectively. The degradation of the RNA sample was analyzed with the DV200 score as a percentage of the total in the region table (%ratio of ≥200 per region). The panel analysis of genetic mutations is becoming less invasive, from the use of tissue specimens to blood tests. Cytological specimens have the advantage of being collected more easily and safely than tissue specimens, but the efficacy of these samples is not sufficiently investigated. Against this backdrop, this report is the first to describe the successful detection of the EML4-ALK fusion gene with the NGS panel analysis using sputum cytology, a non-invasive method. Although sputum cytology is a pathologically established diagnostic method with a sensitivity of 66% and a specificity of 99% [5], the frequency of positive results in lung adenocarcinoma is not high [6,7]. Recently, gene mutation detection using both sputum and digital droplet PCR method was reported [8,9]. Other similar studies of NGS analysis of sputum using ctDNA were also reported [10,11]. We previously described three cases where successful gene panel analysis led to the therapeutic effects of molecular-targeted drugs for EGFR and MET mutations in NSCLC [3]. In this report, the EML4-ALK fusion gene was detected from sputum cytology specimens using RNA analysis. This finding, that very fragile RNA can be detected in sputum specimens, is considered important. This case study has some limitations. First, this is a case report with very little experience in the analysis of NGS in sputum, and although it is an important fact, the results are preliminary. Second, this was a case of an advanced disease with atelectasis, and it is possible that fresh adenocarcinoma cells were likely to be expectorated. It is not known to what degree of bronchial involvement leads to positive sputum cytology or if expectorating viable cells can be detected. Third, there is a large variation in the quality of sputum. Therefore, there are issues to be considered regarding the method of treatment for sputum itself since no standardized principles are established. It is expected that guidelines for panel testing using cytology will include the handling of sputum cytology in the future. LCCP can detect mutations in eight genes with high sensitivity, but if they are all negative, comprehensive genetic analyses such as PD-L1 expression, tumor mutation burden, and other rare gene detections are required. We plan to develop LCCP so that a comprehensive analysis can be performed in the future. Liquid biopsy is also a simple and minimally invasive technique, but its use in the first visit is not covered by medical insurance in our country, so it cannot be used in this case. However, it is hoped that the gene panel test using blood will spread as a simpler method in the future. We write this article in which EML4-ALK was identified by RNA analysis in sputum cytology samples using the new lung cancer NGS panel CDx, LCCP. Although this is a non-invasive option, future prospective studies will be necessary to confirm its versatility.

## Data Availability

https://www.ddbj.nig.ac.jp/index.html (accessed on 10 June 2023); submission: DRA016449 (keimorikawa-0003_Submission); BioProject: PRJDB15968 (PSUB020490); BioSample: SAMD00620266 (SSUB025804); experiment: DRX466817 (keimorikawa-0003_Experiment_0001); run: DRR482591 (keimorikawa-0003_Run_0001).

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
