# Peer review of "EML4-ALK Gene Mutation Detected with New NGS Lung Cancer Panel CDx Using Sputum Cytology in a Case of Advanced NSCLC"

_diagnostics, 2023, doi:10.3390/diagnostics13142327_

Round 1

Reviewer 1 Report

This is a case report showing that sputum cytology is feasible for EML4-ALK rearrangement detection using a newly approved panel – LCCP. Some minor suggestions:

1.     Some details of this patient described in this case report:

a.     Smoking history;

b.     How was the sputum sample taken from the patient, including patient preparation, amount of the sputum, route of the sputum, by patient himself or through the intubation tube?

c.     In Figure 1, how long was the treatment administered when the tumor became smaller?

d.     For the imaging of head in Figure 1, was it CT or MRI?

2.     What are the pros and cons of the LCCP panel using sputum sample when compared with other options, especially using blood samples, which is also very frequently used and easier to be obtained than tissue sample? It would be more comprehensive if the authors can describe some details regarding this in the discussion section.

3.     It looks like the LCCP only includes 8 genes (EGFR, BRAF, KRAS, ERBB2, ALK, ROS1, MET, RET) for lung cancer. In real clinical practice, 8 genes are not enough for decision-making, especially if patients have all negative results. TMB is another prediction marker to immunotherapy for this group of patients. And 8 genes are not sufficient for the TMB detection. It could be some limitations.

Author Response

Answer to reviewer 1

Thank you for pointing out many details that were missing in this paper. I will answer the points from the reviewer one by one.

  1. Some details of this patient described in this case report:

Answer: we added the sentence as below.

  1. Smoking history;

The patient was a 67-year-old man who had never smoked but had hypertension and type 2 diabetes.

  1. How was the sputum sample taken from the patient, including patient preparation, amount of the sputum, route of the sputum, by patient himself or through the intubation tube?

He was extubated on hospital day 4 and transferred to our respiratory medicine department. The next day, the sputum that he coughed up was cytologically positive; classâ…¤adenocarcinoma

  1. In Figure 1, how long was the treatment administered when the tumor became smaller?

Both the primary tumor and brain metastases showed remarkable improvements after one month (figure 1B) with CT assessment.

  1. For the imaging of head in Figure 1, was it CT or MRI?

Same as above sentence.

  1. What are the pros and cons of the LCCP panel using sputum sample when compared with other options, especially using blood samples, which is also very frequently used and easier to be obtained than tissue sample? It would be more comprehensive if the authors can describe some details regarding this in the discussion section.

Answer: We added a discussion as a last part of limitation.

Liquid biopsy is also a simple and minimally invasive technique, but its use at the first visit is not covered by medical insurance in our country, so it cannot be used in this case. However, it is hoped that gene panel test using blood will spread as a simpler method in the future.

  1. It looks like the LCCP only includes 8 genes (EGFR, BRAF, KRAS, ERBB2, ALK, ROS1, MET, RET) for lung cancer. In real clinical practice, 8 genes are not enough for decision-making, especially if patients have all negative results. TMB is another prediction marker to immunotherapy for this group of patients. And 8 genes are not sufficient for the TMB detection. It could be some limitations.

Answer: We added a discussion as a last part of limitation.

LCCP can detect mutations in eight genes with high sensitivity, but if all are negative, comprehensive genetic analysis such as PD-L1 expression, tumor mutation burden, and other rare genes detection are required. We plan to develop LCCP so that comprehensive analysis can be performed in the future.

Reviewer 2 Report

The authors claim to have identified evidence of fusion of EML4 to ALK using nucleic acid recovered from patient sputum analysed by panel sequencing but the basis for calling specifically the EML4 aspect of the study is not established by the data presented - merely the presence of ALK RNA (if I understand the assay correctly).  Additional analysis is necessary to support the claim of fusion between these 2 genes.

As a minor point, please correct the sentence "RNA degrades nucleic acids quickly" which I suspect is an editing mistake

Language is fine.

Author Response

Answer to reviewer 2

Thank you for pointing out many details that were missing in this paper. I will answer the points from the reviewer one by one.

The authors claim to have identified evidence of fusion of EML4 to ALK using nucleic acid recovered from patient sputum analysed by panel sequencing but the basis for calling specifically the EML4 aspect of the study is not established by the data presented - merely the presence of ALK RNA (if I understand the assay correctly).  Additional analysis is necessary to support the claim of fusion between these 2 genes.

Answer: We added the variant information as below.

The detected RNA mutational variants extracted from tissues and sputum matched EML4exon6ins33_ALKexon20.

As a minor point, please correct the sentence "RNA degrades nucleic acids quickly" which I suspect is an editing mistake

Answer: We revised the sentence as below.

However, the quality of RNA tends to degrade quickly, it is generally recommended to use frozen or formalin-fixed paraffin-embedded (FFPE) tissue specimens to secure a sufficient amount of nucleic acids.

Round 2

Reviewer 2 Report

The authors have made the appropriate textual amendments and the manuscript if fine for publication